# The Influence of Corrosion Processes on the Degradation of Concrete Cover

**DOI:** 10.3390/ma17061398

**Published:** 2024-03-19

**Authors:** Zofia Szweda, Artur Skórkowski, Petr Konečný

**Affiliations:** 1Department of Building Structures, Faculty of Civil Engineering, Silesian University of Technology, 44-100 Gliwice, Poland; 2Department of Measurement Science, Electronics and Control, Faculty of Electrical Engineering, Silesian University of Technology, 44-100 Gliwice, Poland; artur.skorkowski@polsl.pl; 3Department of Structural Mechanics, Faculty of Civil Engineering, VSB-Technical University of Ostrava, 70800 Ostrava, Czech Republic; petr.konecny@vsb.cz

**Keywords:** accelerated corrosion, concrete cover, corrosion initiation time, time of activation, mechanical impact, corrosion products, cracking time

## Abstract

In this work, two methods were used to accelerate the corrosion of concrete. In the first method, chloride ions were injected into the concrete using the migration method. The moment of the initiation of the corrosion process was monitored using an electrochemical method of measuring polarization resistance. In the next step, the corrosion process was accelerated by the electrolysis process. Changes on the sample surface were also monitored using a camera. In the second method, the corrosion process of the reinforcing bar was initiated by the use of the electrolysis process only. Here, changes occurring on the surfaces of the tested sample were recorded using two web cameras placed on planes perpendicular to each other. Continuous measurement of the current flowing through the system was carried out in both cases. It was assumed that in conditions of natural corrosion, a crack would occur when the sum of the mass loss of the reinforcing bar due to corrosion reached the same value in $t_{cr(real)}$ (real time) as it reached in the $t_{cr}$ (time of cracking) during the accelerated corrosion test. The real time value was estimated for C1 concrete with cement CEM I. The estimated value was $t_{cr(real)}$ = 1.1 years and for C2 concrete with cement CEM III, $t_{cr(real)}$ = 11.2 years. However, the main difference that was observed during the tests was the nature of the concrete cracks. In the case of the C1 concrete sample, these occurred along the reinforcing bar, while in the C2 concrete, the failures occurred on a perpendicular plane transverse to the direction of the reinforcing bar.

## 1. Introduction

It is assumed that a properly selected thickness of concrete cover provides sufficient protection for reinforced concrete structures. There are documents regulating the methods for designing the thickness of concrete cover for reinforced concrete and prestressed structures, taking into account the type of concrete used and the environmental classes in which the designed objects will be operated [1,2]. The random distribution of pore spaces present in the concrete cover is the reason why aggressive substances such as chloride, carbon dioxide, oxygen, etc., can penetrate through weak points, causing corrosion of steel bar reinforcements in concrete and ultimately inducing concrete cracking [3]. The corrosive products have 1.7–6.2 s of their original iron volume and are usually deposited in the pores of the concrete [4]. Since the volume of pores present at the contact surface between the reinforcing bar and the concrete is limited, the accumulation of corrosive products exerts pressure on the surrounding concrete, causing tangential tensile stresses. When the value of these stresses exceeds the tensile strength of the concrete, cracks will appear. This may subsequently lead to the destruction of the concrete cover [5,6]. It is known, however, that under certain favorable conditions, especially in very wet concrete, some of the corrosive products can be released from the surface of the reinforcing bars, thus partially alleviating the tensile stresses around the reinforcing bar [7]. In order to ensure the stable and safe operation of structures, methods of analyzing and predicting the durability of reinforced concrete structures should be developed [8,9,10]. The service life of structures exposed to aggressive environments may be divided, according to [11], into an initiation period when, e.g., chlorides penetrate to the level of the reinforcement and their amounts exceed the chloride threshold. Then, the corrosion initiates and a propagation period starts. Chloride penetration is the first part of this process and is usually modeled using Ficks’s second law of diffusion [12,13]. Sometimes, finite element models are used for more complicated boundary conditions, e.g., when considering the effect of cracks on chloride ion penetration [14,15,16]. The propagation period is related to the oxidation of steel reinforcements and the formation of rust products, followed by the concrete cover cracking. Usually, cracking of the concrete cover is considered to indicate the end of the service life of concrete structures affected by steel reinforcement corrosion [17]. Since classical methods of examining the development of corrosion processes are time consuming [18,19], various methods of accelerating the corrosion process are used [3,20,21,22,23]. These methods usually use an external electrical field and high corrosion rate values (typically 100 μA/cm^2^ or higher). A typical lollypop reinforced concrete test specimen (named after its shape) and setup was used by Care and Raharinaivo [24] for an accelerated corrosion study using the impressed current technique. Electrolytes containing both chloride ions and pure water were used in the tests. The test results indicated that corrosion processes can be better described by Faraday’s law when using an electrolyte containing chloride ions.

Alternative methods for accelerating the corrosion of steel in concrete have also been reported in the literature. Since corrosion processes accelerated by an electric field proceed differently to those occurring in a natural corrosion environment, Yuan et al. [25] proposed an alternative method of accelerated steel reinforcement corrosion induction using an artificially modeled climatic environment. It was concluded that the artificially modeled environment in the climatic chamber better models the natural processes of corrosion than methods using electric field acceleration. Chunlei et al. [26] proposed a method of accelerating the corrosion of steel reinforcements by accelerating the diffusion of chloride ions into concrete using an electric field. This was in order to achieve a critical value of chloride ion concentration at the surface of the reinforcing steel, causing corrosion initiation, in a shorter time. In a short two-day test, the LPR linear polarization method and direct observation of the rod surface (by breaking the sample) were used to assess corrosion states. Based on the results, the effectiveness of the tested inhibitors was assessed. However, in this method, the conditions favoring corrosion are accelerated, and the corrosion initiation process itself is assessed.

All these studies and observations are the starting point for developing theories and methods for modeling the behavior of reinforced concrete structures under the influence of the development of corrosion processes [27].

The commonly accepted model assumes a perfectly homogeneous nature of concrete and a uniform disintegration of corrosive products around the cross-section of the reinforcing bars. It is expected that the mode of failure depends primarily on the cover depth c and the reinforcing bar spacing s. If c is relatively small in relation to s, cracks inclined at an angle of 45° are likely to occur. When these cracks reach the concrete surface, spalling of the concrete surface will occur. If rebars are placed too closely together, cracks tend to develop and progress in the plane of the rebar, causing them to delaminate. Such cracks may therefore not be visible from the external surface in the initial phase of corrosion [28].

Numerous studies have been focused on the corrosion-induced cracking of concrete, leading to the proposition of various cracking models [5]. These models can be categorized into three main types: empirical (experimental) [29], analytical [30,31,32], and numerical [15,33,34]. Empirical models are often developed with corrosion initiation accelerating methods, such as the application of external impressed currents, resulting in <0.05 mm cracks. Replicability of the empirical models is very poor, especially when it comes to predicting the $t_{cr}$ (years, concrete crack appearance time) [35]. In [36], a new prediction model that takes into account the influence of changes in ambient temperature and humidity, concrete resistivity, chloride ion concentration in concrete, corrosion duration, water–cement ratio, and cover thickness was proposed.

Analytical models are based primarily on assumptions of mechanics of deformable bodies. Several simplifying assumptions are applied in a typical analysis. The concrete encompassing the reinforcing bars is treated as a thick-walled cylinder, and it is regarded as a brittle, isotropic, linearly elastic material. Corrosive products are assumed to be uniformly distributed both around and along the reinforcing bars, concrete stresses along the bar axis are considered negligible, pores and voids at the steel–concrete interface are assumed to be best represented by a uniform empty band encircling the reinforcing bars, and only stresses arising from the pressure exerted by the expanding corrosive products are considered in the model [5]. It is assumed that $t_{cr}$ (corrosion propagation time until the concrete cover cracks) is directly proportional to $w_{cr}$ (g/mm^2^, the critical amount of corrosion products causing tensile stresses greater than the tensile strength of concrete) and inversely proportional to $i_{corr}$ (corrosion rate of reinforcing steel). The parameter $w_{cr}$ is intricate and influenced by various factors, including the type of corrosion products formed, their ultimate deposition locations, the concrete’s porosity, their mobility within the concrete, and material properties such as the tensile strength and modulus of elasticity [5,37].

Finite element modeling (FEM) is the main method used in numerical modeling of corrosive products’ expansion [38,39]. Compared to other models, it allows for more convenient adjustment of material properties and geometric complexities. Still, there are concerns about validating this model, since they are closely related to the simulated object and a specific situation in FEM projects [5,27,40,41]. The work in [17] presents a mechanical model where XFEM-Based Crack Growth Simulation module of Ansys Software is used to describe the distribution of stresses in the cross-section samples in the corrosion process. However, there was a problem with determining the cracking time, since in all cases, shorter times than observed in experimental studies were obtained.

All things considered, there is still a need for more experimental tests to determine the impact of corrosive products on the mechanical properties of steel-reinforced concrete. This requires numerous experiments and many different measurements (mechanical, geometric, electrochemical) on simple structures to analyze the corrosion development, its mechanical effects, and the final products. It is also important to improve the knowledge of the relationship between rate of corrosion and cracking [4,42]. 

In this work, research on the development of corrosion of reinforcing bars was carried out using an electric field to accelerate the process. Two methods were used to accelerate corrosion. In the first method, chloride ions were injected into the concrete using the migration method to initiate the corrosion process, accelerating their penetration by using an electric field. The moment of initiation of the corrosion process was monitored using an electrochemical method of measuring polarization resistance. Additionally, when the corrosion process was detected, the distribution of chloride ion concentrations across the thickness of the concrete cover was determined. In the next step, the corrosion process was accelerated using the electrolysis process, by connecting a constant voltage source of 30 V. At the same time, the current flowing through the system was measured. Changes on the sample surface were monitored with a camera that automatically took pictures at specific intervals.

In the second method, the corrosion process of the reinforcing bar was accelerated using the electrolysis process only. The system consisted, analogously to the previous method, of a reinforcing bar and a rust-resistant steel electrode applied to the external surface of the sample. The source of chloride ions was the electrolyte in the form of a 3% NaCl solution. Here, changes occurring on the surfaces of the tested sample were recorded using two web cameras placed on planes perpendicular to each other. Continuous measurement of the current flowing through the system was carried out. Each method was tested for 24 days. 

The aim of the research was to find the relationship between the value of the current flowing in the system, which, according to Faraday’s law, is proportional to the mass of the corrosion products formed, and the type and width of the cracks in concrete. The research is preliminary and was conducted with the intention to compare the methods in order to select the optimal one to be implemented in further research. Since both tests used concrete of the same composition but with different types of cement, it is also possible to make a preliminary comparison of the cracking resistance of these concretes. 

### Significance and Novelty

From the economic standpoint, the correct and precise determination of the durability of a reinforced concrete structure is essential, since it provides information regarding how long the structure will be used without additional costs for renovation or even demolition. The process of concrete cracking due to the corrosion of reinforcing bars is very complex and depends on many factors such as concrete porosity, concrete composition, type of cement used, type of steel, and the amount of aggressive factors causing corrosion. 

This research project uses an accelerated process of chloride ion penetration into concrete to induce corrosion processes coupled with the recording of these processes by the method of linear concrete polarization (LPR). Thanks to this procedure, the value of the corrosion current velocity occurring in natural conditions for the tested steel–concrete system was obtained and $t_{cr}$—the cracking time of concrete in conditions of accelerated corrosion—was directly determined. Additionally, this method allows for the control of the concentration of chloride ions in the accelerated corrosion process. Accelerated corrosion tests are performed to obtain information used in predicting the durability of structures; however, the results of accelerated tests should not be directly extrapolated to real predictions of the service life of real structures [43]. Moreover, the real predicted time of concrete cracking resulting from accumulation of corrosion products was estimated thanks to the assumption that the crack will occur as a result of the formation of the same amount of corrosion products that caused the cracking of the tested sample in the simulated corrosion test.

The literature on changes in concrete strength due to the admixture of chloride ions is not clear. In some cases, the addition of chloride ions causes slight changes in compressive strength, as shown in [44]. The results of tests on self-compacting concretes presented in [45], in the case of concrete containing Portland cement, indicate a minimal decrease in strength with increasing chloride content. However, in the case of concrete with the addition of fly ash, the compressive strength increases with the addition of chloride ions. At the same time, the use of an electric field and chloride ions can also change the microstructure of concrete by creating ettringite, which, in the initial phase, can contribute to strengthening the microstructure of concrete, and in the later stage, to bursting and weakening of the structure, similar to the process of extracting chloride ions from concrete [46].

In the second method, the entire cracking process was recorded, which, after improving the quality of the recorded image, will allow us to determine the dependence of the mass of corrosion products generated during the process on the appearance and the propagation of this crack. Taking that additional information into account supports a more accurate prediction of corrosion processes. 

## 2. Materials 

The research was carried out with two ordinary concrete types of a comparable composition, differing only in the type of cement from the Lafarge cement plant in Małogoszcz (Poland) used. In concrete C1, CEM I 42.5 R cement was used, whilst in the second concrete (C2), CEM III/A 42.5 N-LH/HSR/NA cement was used. The specimens were prepared and every test was conducted at the Laboratory of Civil Engineering of the Silesian University of Technology. Compressive strength, density, and porosity tests were described in [47] where, among others, C1 and C2 concretes were used in diffusion tests. Properties and compressive strength of the analyzed concrete mixtures from all series are presented in Table 1. 

The detailed chemical composition and basic properties (according to the producer’s specification) of the cements are given in Table 2, and the detailed compositions of concrete mixes C1 and C2 are presented in Table 3. 

The molds were of a height of 5 cm, with 10 cm diameter. The chloride ion penetration was the same for all the samples: vertical from the top of the sample. Ribbed rebars with ø12 mm, made of B500SP steel, were positioned inside the specimens perpendicular to the direction of the cylinder axis, at the center of its cross-section. The applied reinforcement cover was 20 mm wide. The prevention of crevice corrosion on the contact elements of the rebar ends, which were situated on the sides of the cylindrical specimen, was ensured through electrical insulation. These elements were connected to a conductor on one side. Figure 1a shows the specimens prepared for testing, with plastic tanks made of PVC pipes attached to the upper surface of these elements. The same model of specimens was used previously [48] to determine the impact of concrete design on the effectiveness of the electrochemical chloride extraction process. The same model, but without reinforcement, was used [47,49] to determine the values of the diffusion coefficient of chloride ions. These models were used in this research because of the intention to model the processes of chloride ion diffusion in concrete and reinforcement corrosion processes on them in future research.

## 3. Test Methods

The original intention of the research was to compare the influence of the cement used in concrete on crack propagation. That is why samples with different cements were used in both methods of inducing corrosion. However, the conducted tests were preliminary in nature and led to obtaining very different results regarding the type of cracking in the tested methods. Therefore, the new intention of the research was to collect experimental data related to corrosion initiation and crack propagation time in order to further calibrate numerical models. 

In method 1, a voltage of 18 V was used only in the chloride ion charging stage, while in the second stage (electrolysis), the same voltage was used as in method 2 (30 V). In the second method, the electrolysis occurred from the very beginning of the test and the solution in which the samples were soaked during the test was the source of chloride ions. These differences are present due to the fact that both methods are used to accelerate corrosion in the literature, and the aim of the study was to conduct a preliminary analysis in order to select the best method for further research.

### 3.1. Accelerated Corrosion Process Method 1

#### 3.1.1. Accelerated Process of Chloride Migration in Concrete

Four samples made of C1 concrete were used in the test. The chloride migration process was carried out similarly to [46,48,50]. The specimens made of C1 concrete were immersed in water for 72 h ahead of the tests. The first set of specimens (1) was placed on top of a large rectangular electrode (anode), constructed from titanium mesh (coated with a thin layer of platinum) (2) and submerged in tap water at the base of a shallow tank (3). At the top, plastic containers (4) filled with a 3% NaCl solution up to a height of 7 cm were placed. A round stainless steel electrode (cathode) (5) with a diameter adjusted to the tank hole was placed on top of each specimen inside each tank. The test set was supplied with 18 V direct current—Figure 1. 

Before subjecting concrete to accelerated chloride migration through an electric field, polarization tests using the LPR method were conducted on all specimens to ascertain the corrosion potential of reinforcement in its passive state. The chloride migration process was halted every 7 days to observe the corrosion progression by measuring the corrosion potential. Electrochemical measurements were conducted 3 days after discontinuing the electrical supply to prevent polarization of the tested reinforcement [46,48,50].

The measurements followed a three-electrode setup, employing a steel rebar as the working electrode (2). The counter electrode (4), crafted from a stainless steel sheet, was tailored to fit the test specimens’ shapes. A reference electrode (5) of Cl^−^/AgCl,Ag composition was positioned on the cylinder surface, snugly secured to the plastic tank walls affixed to the specimen. Subsequently, LPR tests were conducted using the Gamry Reference 600 Potentiostat (6) by Gamry Instruments, Warminster, PA, USA (Figure 2a,b) [47,48,51].

After performing two series of charging by the migration method and consecutive LPR measurements, the profile of chloride ion concentration in concrete was determined in one of the tested samples (after 2 weeks of migration). Another chloride ion concentration profile was determined after two further series of charging by the migration method (after 4 weeks of migration). The “Profile Grinding Kit” from German Instruments was employed for this purpose. Concrete was extracted in 4 mm thick layers at 10 consecutive levels (Figure 2c). Solutions were obtained from the crushed concrete. The concentration of chloride ions and pH in these solutions were determined using a multi-functional multimeter from Elmetron.

For a detailed description of the experimental and calculation procedure used to determine ${\bar{C}}_{t\;exp},$ refer to the articles [46,47,48,49,50,51]. 

#### 3.1.2. Accelerated Process of Corrosion 

After introducing chloride ions into the concrete to initiate corrosion processes, an electric field was applied to expedite the speed of these processes. The accelerated corrosion test utilized a potentiostat (Silesian University of Technology, Gliwice, Poland), designed to induce corrosion in the reinforcement rebar. In this setup, the reinforcing bar (1) served as the anode, while the cathode was composed of rust-resistant perforated steel sheet (2). Both electrodes were connected to the power supply (potentiostat) (3) using insulated wires. The potentiostat unit (3) facilitated the automatic recording of fundamental electrical parameters, including current (I) and voltage (U), at a preset frequency of 60 s. A constant voltage of 30 V was maintained throughout the entire testing period. The concrete sample (4) was immersed in tap water (5) to ensure electrical contact for all electrodes in the system. Throughout the study, changes in the sample’s surface were documented using a camera (6), capturing images at 3-day intervals—Figure 3. 

The employment of direct current markedly expedites the experimental timeline. The applied voltage induces a substantial shift in the potential of the steel electrode, with iron consumption emerging as the predominant process: Fe − 2e^−^ = Fe^2+^.(1)

Taking into account the dominant process, the determination of the mass of oxidizing iron ions was derived through the application of Faraday’s law [52]:(2)$$∆m(t)=\int_{0}^{t}kIdt,\;\;\;k=\frac{M}{zF}$$
where ∆m(t) is the mass of iron ions transported from the steel rebar into the concrete microstructure after an electrolysis duration of time ‘*t*’, where *‘k’* denotes the electrochemical equivalent. It is noteworthy that the current intensity is not specifically associated with the charge flow within the corrosion microcells; rather, it signifies the total intensity of the external current denoted by ‘*I*’. The variables ‘*M*’ and ‘*F*’ represent the molar mass of iron (56 g/mol) and the Faraday constant (*F* = 9.6485 × 10^4^ C/mol), respectively. Additionally, ‘*z*’ indicates the number of electrons involved in the oxidation reaction, with values of 2 or 3.

The indiscriminate calculation of the coefficient *k* equal to *k* = 1.93 × 10^−4^ g/C and including it in Formula (4) may lead to large discrepancies between the actual mass of iron ions transferred to concrete microstructure measured gravimetrically and that calculated from the formula [53]. This is because electrolysis does not take place in a pure solution, and other ions present in the liquid, besides iron, take part in the process of transferring the electric charge. Auyeung et al. [54] discovered disparities between theoretical and observed corrosion mass loss. These differences can be attributed to multiple factors, including the requirement of electrical energy to initiate corrosion, the resistivity of concrete, the composition of the reinforcement bar, and the electrical properties of minerals present in the concrete.

#### 3.1.3. Measurement of the Crack Opening Width Using the Optical Method

The surface of the sample was covered with white paint and a pattern of small black dots before the tests. During the experiment, photos of the sample surface were taken at intervals of 3 days. Then, the obtained images were analyzed using the Gom correlate program, which was used to measure the width of the crack opening. In this method, a grid of analyzed points is created, dividing the analyzed area. The first photo is taken before any deformations occur, then a series of subsequent photos show the samples subjected to a load of corrosive products. Those products cause tensile stresses in the cover when their volume increases and subsequently cause its cracking. A series of photos shows the consecutive stages of deformation caused by the acting load through the set of displacement values of marked points on the surface of the analyzed sample. The system reads changes in the position of points by registering their coordinates.

### 3.2. Accelerated Corrosion Process Method 2

One sample made of C2 concrete was used for testing. A concrete sample (1) made of C2 concrete containing a metal reinforcing bar (2) was placed on a titanium mesh (3) in a container (4) with a mixture of water and salt (3% NaCl salt concentration) (5) in a way that prevented it from coming into contact with the bar. Electric wires (6) were connected to the end of the titanium mesh covered with a layer of platinum and to the metal reinforcing bar, connected to a power source (potentiometer) (7) with a voltage of 30 V. The electrolysis carried out in this way was intended to lead to a faster corrosion process of the reinforcing bar, resulting in the cracking of the sample. The entire experiment lasted about 3 weeks, and the sample monitoring process was carried out using two webcams (one placed in a top view (8) and the other directed at the place of the expected sample crack (9)). The OBS program was used to operate the cameras, allowing for a transmission of views from both cameras on one screen. The sample was illuminated with a lamp to maintain a constant light level at all times. Due to the duration of the experiment, it was necessary to supervise the whole experimental setup every day in order to replenish the water that had evaporated and check the recording process to determine whether any errors occurred.

During the experiment, the measurement system recorded changes in the electric current flowing between electrodes 2 and 3—Figure 4. 

## 4. Results and Discussion

### 4.1. Results of the Accelerated Corrosion Process—Method 1

#### 4.1.1. Result of the Accelerated Process of Chloride Migration in Concrete

Chloride concentration and pH levels were assessed in pore solutions made of ten 4 mm layers of grinded concrete. The results of tests conducted on two samples, namely C1.1 after 14 days and C1.2 after 28 days of chloride ion migration in concrete, are depicted in Figure 5. This figure illustrates the distribution of chloride concentrations and pH values with respect to the depth of the reinforcement cover.

As depicted in Figure 5a, the chloride concentration in the C1.1 concrete sample, near the reinforcing bars, was approximately 1.4% of the cement’s weight after 14 days of migration. Similarly, in the C1.2 concrete sample, near the reinforcing bars, the chloride concentration exceeded approximately 1.4% of the weight of cement after 28 days of migration. Notably, the concentration of chloride ions surpassed the critical value across the entire depth of the concrete cover in both C1.1 and C1.2 samples.

In accordance with the standard criterion [2], the likelihood of reinforcement corrosion appeared to be imminent in both cases. The measurements of reinforcement polarization substantiated this presumption, revealing elevated corrosion current values after 7 days, which persisted and were further validated after 14 days of chloride migration into the concrete across all tested samples. Subsequent assessments of the corrosion current, conducted after 21 and 28 days of migration, continued to affirm the sustained presence of the corrosion process at a moderate level in samples C1.3 and C1.4.

By analyzing the carbonation of concrete based on the pH value, it can be concluded that the steel in concrete is not exposed to corrosion (Figure 5b). Considering the added presence of hydroxide ions (Figure 5b), an evaluation using the Hausmann criterion (Figure 5c) reveals that, based on the corrosion current measurements in samples C1.1 and C1.2, the migration process of chloride ions appears secure and should not escalate to a state of corrosion risk. However, this assumption was not substantiated by corrosion measurements. A markedly different interpretation arises when examining the results concerning the corrosion risk limit value for the ratio [Cl^−^]/[OH^−^] ≤ 0.1. For the set of results the graphs are based on, refer to the dataset [55].

As a result of LPR tests, polarization curves (Figure 6), where the horizontal axis shows the relationship between the current signal and the vertical axis shows potential, are obtained. The polarization process starts from the negative direction, i.e., from reducing the potential value, thanks to which a cathodic reaction polarization curve is obtained, which is equivalent to the kinetics of the reduction reaction. Then, reinforcement is polarized in the positive direction, obtaining an anodic polarization curve, which is equivalent to oxidation reaction kinetics. The outcomes of these tests yield the values of corrosive current densities, providing a distinct measure of the corrosion rate for the reinforcement. The corrosion current $(i_{corr}$(μA)) can be calculated using the polarization resistance ($R_{p}$(kΩ)) obtained through LPR measurement, as per to the Stern–Geary equations [56]: (3)$$R_{p}={\left.\frac{dE}{di}\right|}_{i\to 0,\;E\to E_{corr}},\;i_{corr}=\frac{b_{a}b_{c}}{{2.303R}_{p}(b_{a}+b_{c})},$$
where $b_{a}$ and $b_{c}$ are constants of anodic and cathodic reactions, respectively, coefficients of rectilinear slope for segments of polarization curves—anodic $b_{a}$ and cathodic$\;b_{c}$. For the set of results the graphs are based on, refer to the dataset [55].

The corrosion current density clearly determines the corrosion intensity of steel, since, according to Faraday’s law, the mass of losses ($∆m(t_{cr\left(real\right)})D(mg))$ is proportional to the flowing current ($I_{corr(real)}($μA/cm^2^)):(4)$$D∆m(t_{cr\left(real\right)})=ki_{corr\left(real\right)}t,I_{corr(real)}=\frac{i_{corr(real)}}{S},$$
where *k* is electrochemical equivalent, *t* is time, and *S* is the surface area of the reinforcing bar. The mentioned correlation illustrates the connection between the corrosion current density and the linear corrosion rate ($V_{r}$ (mm/year), expressed as follows:(5)$$V_{r}=0.01159\cdot I_{corr(real)}.$$

The corrosion rate ($V_{r}$ (mm/year) is derived by calculating the average cross-section loss around the circumference of the bar, measured in mm, for each operational year of the structure. 

LPR research was carried out on four research elements. Two elements (C1.1 and C1.2) were excluded from corrosion tests after 2 weeks of migration and used for the determination of the chloride ion concentration distribution in the cross-section of their concrete covers. These samples were destroyed during this test. The remaining two samples (C1.3 and C1.4) were tested for 28 weeks of migration. Comparisons of the results obtained from the analysis of polarization curves for selected samples (C1.1, C1.2, C1.3, C1.4) measured before (M0) and after 7 (M1), 14 (M2), 21 (M3), and 28 (M4) days of chloride migration are shown in Appendix A (Table A1).

During the entire research period, a total of 16 polarization curves were obtained, the shapes of which are shown in Figure 6. 

Figure 7a shows a comparison of the results of successive measurements depicting the corrosion current density $i_{corr}$ of steel reinforcement in concrete across four test elements constructed with C1 concrete. In Figure 7b, a similar comparison is shown, focusing on successive measurements of the corrosion potential $E_{corr}$ of steel reinforcement in concrete from four elements composed of the tested concretes. Considering the conditions outlined in [29,30], the initial reference measurement conducted prior to migration indicated that both corrosion potential (${\bar{E}}_{corr}=28.92\;$< 443 mV) and corrosion current intensity ($i_{corr}$ = 0.05 < 0.3 µA) suggested a passive state of all test elements. Subsequent measurements, taken after 7 days of accelerated chloride ion migration under the influence of an electric field and 3 days post-system deactivation, signaled the initiation of corrosion in all four specimens, as evidenced by alterations in corrosion potential values and corrosion current intensity. Notably, there was a discernible increase in the average corrosion current intensity (${\bar{∆i}}_{corr}$ = 9.57 µA), indicative of a moderate corrosion level. Additionally, the average corrosion potential exhibited a substantial rise, amounting to ${\bar{∆E}}_{corr}$ = 551.84 mV, suggesting a 95% corrosion rate. Following an additional 7-day exposure to chloride ions, a slight decrease in corrosion current intensity was observed ${\bar{∆i}}_{corr}$ = 2.32 µA, while the corrosion potential showed an increase ${\bar{∆E}}_{corr}$ = 77.50 mV. Subsequently, after another 7 days of chloride ion charging (with an additional 3-day period for restraining rebars), a control polarization measurement was conducted. The values obtained from this measurement indicated the presence of corrosion in all the specimens. It can be observed that an increase in corrosion current density for M1 is followed by a decrease for M2, and then the current increases again for M3 and M4. This phenomenon is hard to explain but may perhaps be attributed to a situation where the corrosion products at a given time of measurement (measurement of M2) seal the steel–concrete contact zone, while subsequent measurements of the current are again higher due to diffusion of the corrosion products into the concrete cover. For this reason, several measurements were conducted at certain intervals, as only one measurement would be unreliable. 

#### 4.1.2. Result of the Accelerated Process of Corrosion

The measurement results were graphically depicted to illustrate the progression of changes in current across the analyzed samples. These measurements are presented in Figure 8, showing the changes observed during the test in concrete C1 (Figure 8a) and concrete C2 (Figure 8b). The graph shows characteristic points where sudden changes in current intensity occurred (usually a sudden increase). These points may suggest the time at which the cracking of the sample occurred. For the set of results the graphs are based on, refer to the dataset [55].

Figure 9 shows photos of samples taken close to the times when the current’s increase occurred.

Figure 10 below shows photos taken at characteristic moments (P4, P6, P7) of the surface of the sample made of C1 concrete, analyzed using the GOM Correlate Pro program from Zeiss. The photos show the formation of vertical cracks. 

Figure 11 presents camera shots recorded at characteristic moments (P8, P9, P10, P11, P12, P13) of changes in the current flowing through the sample made of C2 concrete. A particularly visible change in the current intensity occurred at the P12 time mark (after approximately 500 h), which corresponds with the formation of a horizontal crack in the sample, as shown in Figure 11e. It is worth noting that the cracks formed in the C1 concrete samples were vertical.

Table 4 contains the estimated amounts of bar mass loss calculated using the Faraday Equation (4) at characteristic moments (P1, P2, P3, P4, P5, P6, P7) of changes in the current in the C1 concrete sample and at characteristic moments (P8, P9, P10, P11, P12, P13) of changes in the current in the C2 concrete sample. 

In the first method, characteristic points were selected based on photographs taken at certain intervals, apart from point P3, which was selected based on the analysis of the current intensity graph. Abrupt shift in this graph may indicate the point at which cracking begins at the point of contact between the reinforcing bar and the concrete. However, as can be seen in point P4, the crack is slightly visible on the surface of the concrete sample. Thanks to the image analysis performed with the GOM Correlate Pro program from Zeiss, this crack can be determined to be 0.16 mm.

In the second method, characteristic points were selected based on abrupt shifts appearing on the current intensity diagram. Thanks to the continuous image recording, it was possible to obtain an image of the sample at each point. Unfortunately, the quality of these frames and the convex shape of the plane on which the crack occurred do not allow the use of a graphics program to precisely determine the width of the initial crack.

The observations indicate that the location and type of concrete cracks depend not only on the thickness of the concrete cover, but also on the type of cement, the concentration of chloride ions in the concrete [57], as well as the change in the concrete’s microstructure that results from the action of an electric field [58], similarly to the long extraction process [59]. 

It can also be assumed that the value of the time after which cracks may occur is influenced not only by the strength of the concrete, but also by the type of cement used in the tested concrete. However, to confirm this statement, tests of concretes with different cements and the same test methods should be performed.

The tests show that the use of the CEM III cement may contribute to a significant delay in the occurrence of cracking in concrete. Nevertheless, after its occurrence, the propagation of the crack width is faster than in the concrete made with the CEM I cement. Moreover, the tests indicate that the use of the CEM III cement in concrete may also have an impact on the location of the crack in the concrete element.

Figure 12 shows the structure degradation pattern of concrete samples C1 (in P1, …, P7) and C2 (in P8, …, P13) depicted as a mass loss over time, where $t_{in}$ is the initiation time of reinforcement corrosion, $t_{0}$ is the activation time of the mechanical impact of corrosion products, and $t_{cr}$ is the cracking time. It should be noted that the mass loss estimated from Formula (2) is very approximate. Research has shown that the weight loss calculated using this formula is not confirmed by gravimetric tests. The discrepancies in these measurements range from 2% [60] to 5%, 25%, 35% [21,27], and even to 54% [52]. These inaccuracies may result from the fact that in this case, the electrolysis process takes place in concrete, not a solution, and the value of the measured current may be influenced not only by the content of chloride ions in the solution but also by the chemical composition of the concrete material. In concrete containing CEM III cement, even in the initial period of the test, a value six times higher of the measured current can be observed in points P3 and P10—Figure 11. In the subsequent stages of the test, the value of this intensity increases even faster, reaching, in the final stage, a value seven times higher in points P7 and P13—Figure 11. According to Formula (2), it can be concluded that the mass loss of the reinforcing bar is greater in the sample made of C2 concrete than in the sample made of C1 concrete. For the set of results the graphs are based on, refer to the dataset [55].

In the case of the C1 concrete sample tested using method 1, precise determination of $t_{0}$ is possible, since the corrosion processes were induced and detected using the polarization method, before being exposed to an accelerated corrosion process. However, in the case of the second method, it is not possible to isolate the time $t_{in}$ after which the corrosion processes are initiated.

Figure 13 shows a diagram of the degradation of the structure over time, expressed as Δm, the loss of reinforcing bar mass due to corrosion, and $w_{cr}$, the width of the concrete crack opening: (a) C1 concrete samples (in P1, …, P7); (a) C2 concrete samples (in P8, …, P13). A clear relationship can be observed between the progressive loss of mass of reinforcing bars over time and the crack opening width in both C1 and C2 concretes. For the set of results the graphs are based on, refer to the dataset [55].

The $t_{cr}$ time value determined in the tests (defined as the beginning of concrete cracking) applies to the accelerated test. In order to estimate the real time $t_{cr(real)}$, Formula (4) was used, assuming the real corrosion rate of $I_{corr(real)}$ = 10.47 μA/cm^2^ (the maximum value of the corrosion current estimated in point 4.1 by the LPR linear polarization measurement method). 

It was assumed that under conditions of natural corrosion, cracking will occur after${\;t}_{cr(real)}\;$, the time when delta $∆m(t_{cr\left(real\right)})$, the sum of the mass loss of the reinforcing bar due to natural corrosion, reaches the same value as $∆m(t_{cr})$, the sum of the mass loss of the bar obtained in $t_{cr}$, the cracking time during the accelerated corrosion test:(6)$$∆m(t_{cr})=∆m(t_{cr\left(real\right)}).$$

In order to determine the value of the sum of the mass loss of the reinforcing bar, Formula (4) was used, and after taking into account the above assumption (5), the following quotient was obtained:(7)$$k\cdot I_{cr}\cdot {t_{cr}=k\cdot S\cdot I_{corr(real)}t}_{cr(real)}$$

The actual $t_{cr(real)}$ value was estimated from the following formula:(8)$$t_{cr(real)}=\frac{I_{cr}\cdot t_{cr}}{S\cdot I_{corr(real)}}$$
where $I_{cr}\;$(A) is the total intensity of the external current flowing, determined during $t_{cr}\;$(year), and *S* (22.7 cm^2^) is the surface area of the reinforcing bar. 

In order to estimate the real time $t_{cr(real)}$, Formula (4) was used, assuming the real corrosion rate of $I_{corr(real)}$ = 10.47 μA/cm^2^ (the maximum value of the corrosion current estimated in point 4.1 by the LPR linear polarization measurement method). However, $I_{cr}t_{cr}=360.66$ A·s was determined by calculating the area under the curve determined during the accelerated corrosion test (Figure 8a) after the time at which the sample cracked, $t_{cr}$ = 144 h for C1 concrete. Similarly, for C2 concrete, $(I_{cr}t_{cr}=83711.40\;$A·s) was determined by calculating the area under the curve determined during the accelerated corrosion test (Figure 8b) after the time at which the sample cracked, $t_{cr}$ = 328 h. For C1 concrete, the estimated value was $t_{cr(real)}$ = 1.1 years, and for C2 concrete, $t_{cr(real)}$= 11.2 years. It should be noted, however, that in the case of C2 concrete, the time $t_{cr}$ consists of the sum of time $t_{in}$, the reinforcement corrosion initiation, and $t_{0}$, the time of activation of mechanical impact by corrosion products, which cannot be separated in the research method used.

## 5. Conclusions

Based on the conducted tests, the following conclusions can be drawn:Preliminary tests carried out using the accelerated electric field penetration of chloride ions into concrete to initiate corrosion and electrolysis to accelerate corrosion (first method in this work) allow for the estimation of the real time $t_{cr(real)}\;$for the formation of concrete cracks.Based on the tests of the accelerated corrosion process (the second method in this study), it can be inferred that it is possible to estimate the real time $t_{cr(real)}\;$for the formation of concrete cracks. However, it should be remembered that this time consists of the sum of $t_{in}$, the time after which corrosion can be initiated, and $t_{0}$, the time of mechanical impact on the cover concrete.

By comparing the research methods used in these studies, it can be concluded that:The first method used, although more labor-intensive, allows for better control of the concentration of chloride ions contained in concrete, which can have a significant impact on the change in the mechanical properties of concrete.In the first method, more precise determination of the value of the corrosion current occurring in the natural corrosion process, depending on the concentration of chloride ions in the concrete and the type of materials used, is possible.Lack of continuous image measurement during the examination of the first method and the fact that the obtained images do not necessarily coincide with the time read from the current intensity graph based on the disturbances occurring in this graph were disadvantageous for this method.However, in the second method, a continuous image was obtained thanks to the use of a camera, but the quality of the obtained images is not sufficient for image analysis with the Gom correlate program. It is necessary to improve the method of recording image changes during the test by using better-quality cameras.

## Figures and Tables

**Figure 1 materials-17-01398-f001:**
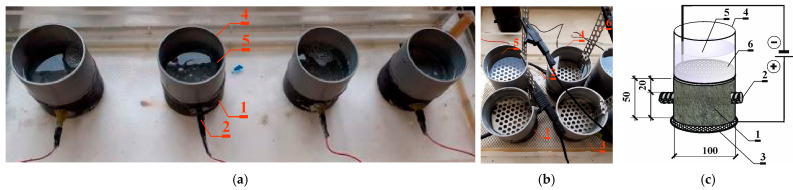
(**a**) Samples prepared for testing from C1 concrete. The experimental setup for accelerating the migration of chloride ions to concrete through the application of an electric field. (**b**) Research view. (**c**) Schematic image of the study: 1—concrete test specimen, 2—ribbed rebar ø12 mm made of steel B500S, 3—titanic anode coated with platinum, 4—small plastic tanks with 5—3% NaCl, 6—stainless steel cathode.

**Figure 2 materials-17-01398-f002:**
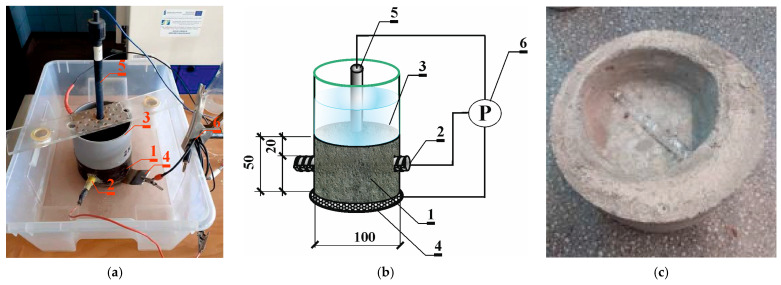
The applied test stand for polarization tests with the LPR and EIS methods: 1—concrete test specimen, 2—ribbed rebar ø12 mm made of steel B500S (working electrode), 3—plastic tank, 4—auxiliary electrode, 5—(Cl^–^/AgCl,Ag) electrode as the reference electrode, 6—Gamry Reference 600 potentiostat with a computer unit and Gamry software: (**a**) photos of the testing procedure; (**b**) scheme of the testing procedure; (**c**) exemplary specimen from which concrete was collected by layers with Profile Grinding Kit.

**Figure 3 materials-17-01398-f003:**
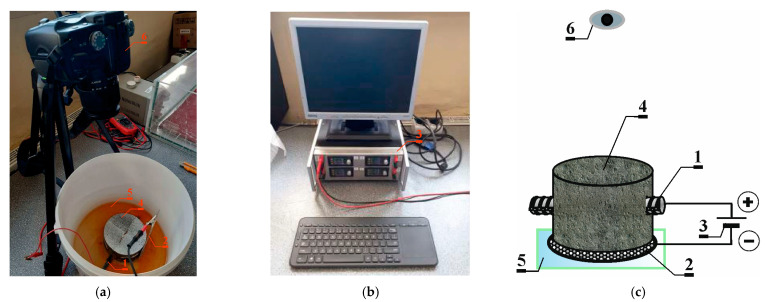
(**a**) Stand for testing accelerated corrosion of the reinforcement with the use of an electric field: 1—reinforcing bar, 2—rust-resistant perforated steel sheet, 3—power supply (potentiostat), 4—sample, 5—tap water, 6—photo camera: (**b**) potentiostat; (**c**) scheme of stand for testing accelerated corrosion.

**Figure 4 materials-17-01398-f004:**
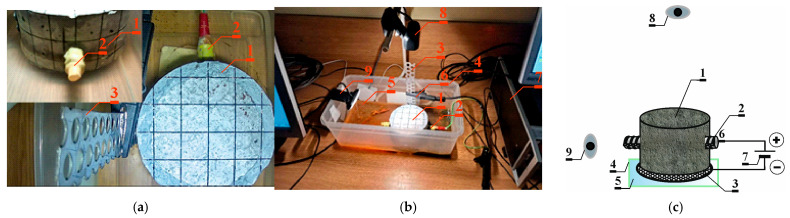
Experimental setup for accelerated corrosion process method 2. (**a**) View of the tested sample; (**b**) stand for testing accelerated corrosion of the reinforcement with the use of an electric field: 1—sample, 2—reinforcing bar, 3—rust-resistant perforated steel sheet, 4—container, 5—3% NaCl salt concentration, 6—electric wires, 7—power supply (potentiostat), 8—webcam placed in a top view of the sample, 9—webcam placed directed at the place of the expected sample crack; (**c**) scheme of stand for testing accelerated corrosion.

**Figure 5 materials-17-01398-f005:**
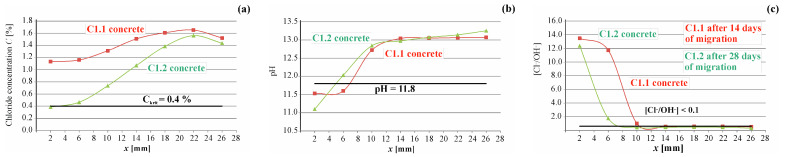
Test results obtained for the reinforced concrete sample C1.1 after 14 days and C1.2 after 28 days of chloride ion migration: (**a**) chloride concentration profiles; (**b**) pH value distribution; and (**c**) distributions in the x direction of the thickness of the concrete cover, the values of the concentration ratios of chloride and hydroxide ions—the Hausman criterion.

**Figure 6 materials-17-01398-f006:**
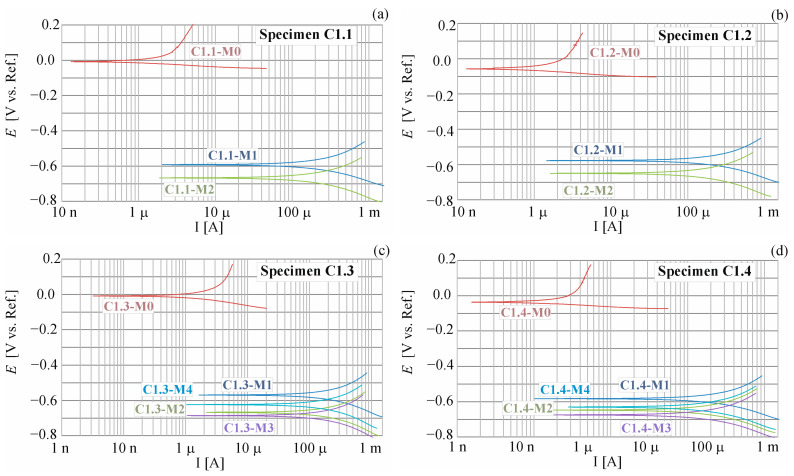
Potentiodynamic polarization curves for steel reinforcement in concrete C1 obtained for selected specimens: (**a**) C1.1, (**b**) C1.2, (**c**) C1.3, and (**d**) C1.4; M0 before chloride migration, M1 after 7 days, M2 after 14 days of chloride migration.

**Figure 7 materials-17-01398-f007:**
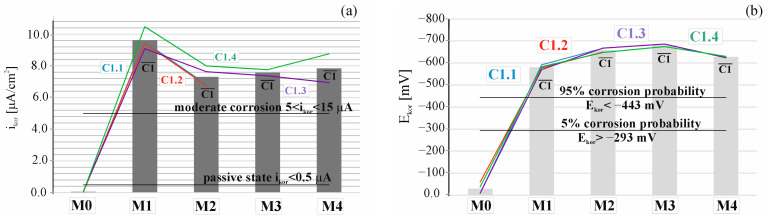
Distribution (**a**) of corrosion current densities and (**b**) corrosion potential obtained for the selected specimens C1.1, C1.2, C1.3, and C1.4: M0—before chloride migration, M1—after 7 days, M2—after 14 days, M3—after 21 days, and M4—after 28 days of migration.

**Figure 8 materials-17-01398-f008:**
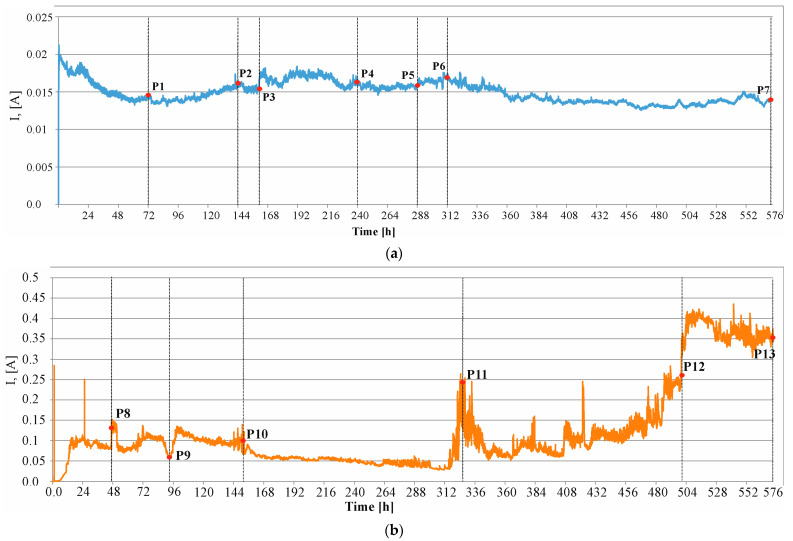
Results of the measurement of current I carried out during the test (**a**) in concrete C1, (**b**) in concrete C2.

**Figure 9 materials-17-01398-f009:**
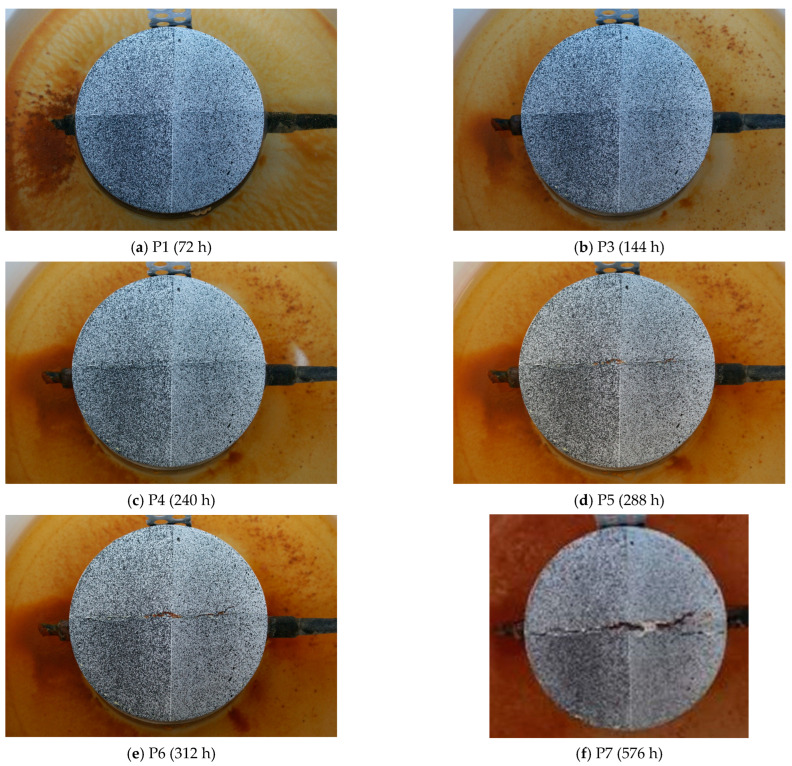
Photos of the surface of the C1 concrete sample taken in the following hours from the beginning of the test: (**a**) P1 after 72 h, (**b**) P3 after 96 h, (**c**) P4 after 96 h, (**d**) P5 after 96 h, (**e**) P6 after 96 h, (**f**) P7 after 96 h.

**Figure 10 materials-17-01398-f010:**
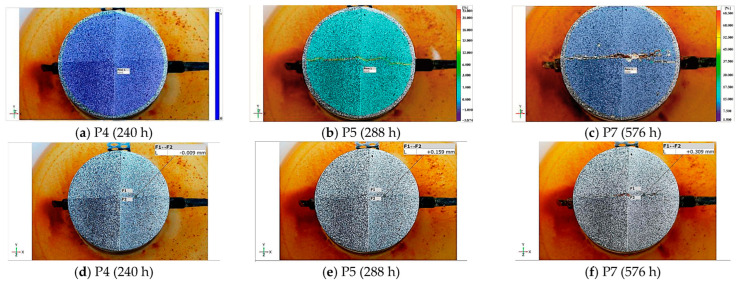
Photos of the surface of the C1 concrete sample taken in the following hours from the beginning of the test analyzed using the GOM Correlate Pro program from Zeiss: (**a**) principal deformation map—P4 after 240 h, (**b**) principal deformation map—P5 after 288 h, (**c**) principal deformation map—P7 after 576 h, (**d**) crack width—P4 after 240 h, (**e**) crack width—P5 after 288 h, (**f**) crack width—P7 after 576 h.

**Figure 11 materials-17-01398-f011:**
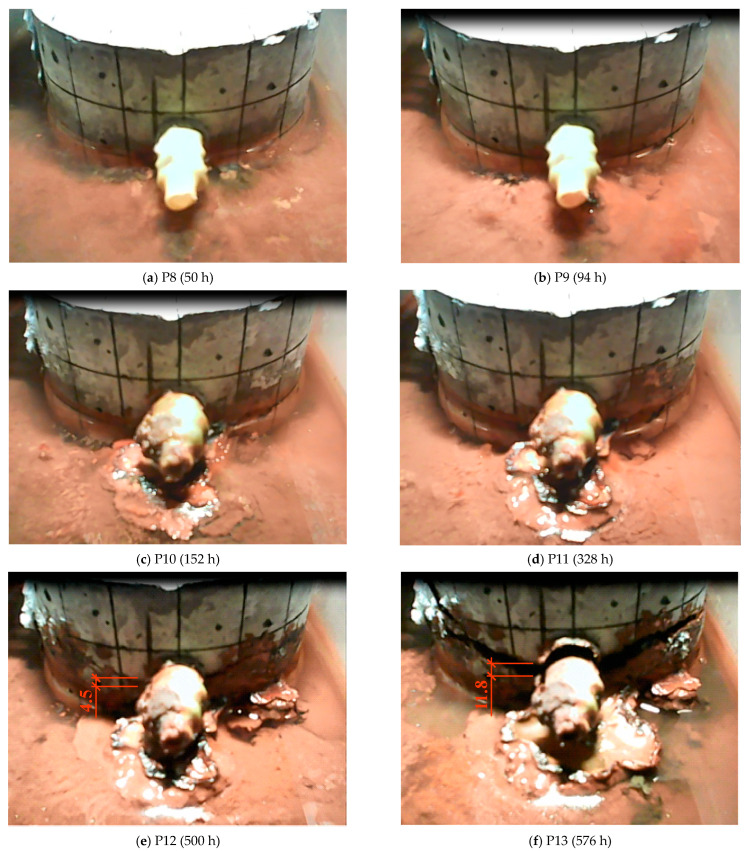
Camera shots recorded at characteristic moments of changes in the current flowing through the sample made of C2 concrete: (**a**) P8 after 50 h, (**b**) P9 after 94 h, (**c**) P10 after 152 h, (**d**) P11 after 328 h, (**e**) P12 after 500 h, (**f**) P13 after 576 h.

**Figure 12 materials-17-01398-f012:**
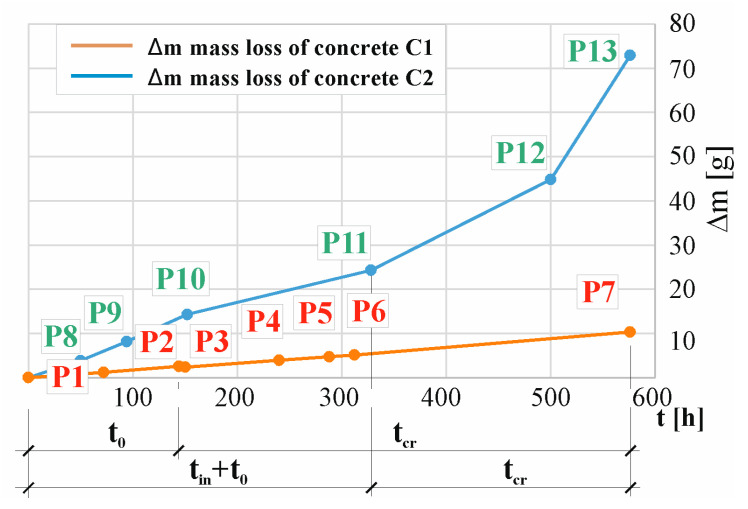
Schematic representation depicting degradation of the structure over time of the C1 (at P1, …, P7) concrete sample and C2 (at P8, …, P13) concrete sample, where $t_{in}$ is the reinforcement corrosion initiation time, $t_{0}$ is the time of activation of mechanical impact by corrosion products, and $t_{cr}$ is the cracking time.

**Figure 13 materials-17-01398-f013:**
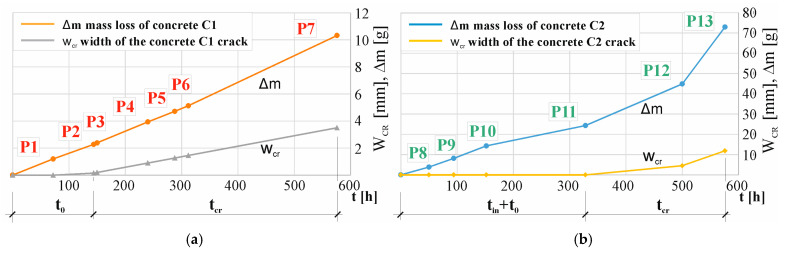
Scheme of structure degradation over time, taken in the following hours from the beginning of the test, expressed as Δm—the loss of reinforcing bar mass due to corrosion and w_cr_—the width of the concrete crack opening: (**a**) C1 concrete samples (in P1, …, P7), (**b**) C2 concrete samples (in P8, …, P13).

**Table 1 materials-17-01398-t001:** Properties and compressive strength of analyzed concrete mixtures from all series.

No.	Compressive Strength [MPa]	Density [kg/m^3^]	Porosity [%]
C1	54.2	2271	12
C2	49.5	2269	7

**Table 2 materials-17-01398-t002:** Chemical compositions of CEM I 42.5 R(C1) and CEM III/A 42.5 N-LH/HSR/NA(C2).

Constituent % mass	Concrete	SiO_2_	Al_2_O_3_	Fe_2_O_3_	CaO	MgO	K_2_O	Na_2_O	Eq. Na_2_O	SO_3_	Cl
	C1	19.38	4.57	3.59	63.78	1.38	0.58	0.21	0.59	3.26	0.069
	C2	29.08	6.30	1.37	48.82	4.36	0.73	0.34	0.82	2.74	0.066

**Table 3 materials-17-01398-t003:** Composition of studied concrete mixtures.

Mixture ID.	Sand (0–2) * mm [kg/m^3^]	Gravel (2–8) * mm [kg/m^3^]	Gravel (8–16) * mm [kg/m^3^]	Type of Cement	Cement [kg/m^3^]	w/c
C1	722	512	2271	CEM I 42.5 R *	681	0.3
C2				CEM III/A 42.5 N-LH/HSR/NA *		

* CEM I—Portland cement; CEM III—blast cement; CEM III/A (65)—maximum content of nonclinker principal components (%); R—high-strength early cement grade; N—grade with normal early cement strength; NA—low-alkali cement; HSR—sulfate-resistant cement; LH—with low heat of hydration.

**Table 4 materials-17-01398-t004:** Bar mass loss ∆m [g] based on Formula (6) and crack w_cr_ [mm] after electrolysis time.

P1 (72 h)	P2 (144 h)	P3 (150 h)	P4 (240 h)	P5 (288 h)	P6 (312 h)	P7 (576 h)
1.19 (0.0)	2.27 (0.11)	2.38 (0.16)	3.93 (0.89)	4.71 (1.27)	5.12 (1.45)	10.32 (3.55)
P8 (50 h)	P19 (94 h)	P10 (152 h)	P11 (328 h)	P12 (500 h)	P13 (576 h)	-
3.83 (0.0)	8.14 (0.0)	14.27 (0.0)	24.28 (0.0)	44.82 (4.5)	72.95 (11.84)	-

## Data Availability

The data presented in this study are available on Zenodo 2022 https://zenodo.org/records/10819340 [55].

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
