# Peer review of "The Influence of Corrosion Processes on the Degradation of Concrete Cover"

_materials, 2024, doi:10.3390/ma17061398_

Round 1

Reviewer 1 Report

Comments and Suggestions for Authors

The authors conducted a study entitled "The influence of corrosion processes on the type of concrete destruction." Although the study presents interesting results, it needs to be improved in order to be accepted for publication.

Please add the cover depth c and the reinforcing bar spacing s to Figure 1. In Figure 1, what do the nomenclatures a and b mean?

Figure 1 should be referred to as (a), (b), (c) and (d), as well as adding more information to the caption.

Authors should describe how samples C1 and C2 were obtained, or at least provide a reference that describes how samples were obtained.

Describe how the compressive strength, density, and porosity tests were conducted, as well as the standard used.

Describe the process by which the authors determined the chemical composition of the two samples. References should be provided.

On page 7, should not Figure 1a be Figure 2a? This is due to the fact that Figure 1 was mentioned on page 3. It is necessary to renumber all figures starting with Figure 1 (on page 3), both in the captions and in the text logo.

Are the references presented by the authors at the end of the sentence "The chloride migration process was carried out similarly to the works" really correct? Please review.

All figures will need to be improved in terms of resolution, some are of poor quality.

Is there a reason why the voltage used in method 2 (30 V) differed from that used in method 1 (18 V)? Can you provide a justification for this?

A dc electric field was applied to the samples in order to conduct the tests. Wouldn't it have been more beneficial to have conducted the study under an ac electric field using the technique of impedance spectroscopy? The impedance technique has been used, for example, by Bortoletto et al. [1], to study the dissolution processes and the period following the maximum conductivity value of Portland cement pastes.

[1] https://doi.org/10.1016/j.conbuildmat.2023.133566

Is there any reason why SEM analyses were not performed to evaluate the microstructure of the samples after corrosion?

In the text, it is unclear why after the increase in corrosion current density for M1 there is a decrease for M2 and then the current increases again for M3 and M4. I would appreciate a more detailed explanation.

In order to enrich the discussion, it is necessary to compare your findings with previous studies and also with those in the literature.

Could you please explain why the accelerated process of chloride migration in concrete tests were not conducted for samples C2 in the same way as shown in Figures 5 and 6 for samples C1?

Overall, the results are very interesting and after corrections, the paper may be published.

Comments on the Quality of English Language

A systematic review of the manuscript is necessary in order to eliminate errors and typos.

Author Response

Response to Reviewer 1 Comments

1) Please add the cover depth c and the reinforcing bar spacing s to Figure 1. In Figure 1, what do the nomenclatures a and b mean?

Ad 1) Figure 1 has been removed because of reviewer #2's recommendation to shorten the introduction. This figure is included in publication no [5].

2) Figure 1 should be referred to as (a), (b), (c) and (d), as well as adding more information to the caption.

Ad 2) Figure 1 has been removed because of reviewer #2's recommendation to shorten the introduction. This figure is included in publication no [5].

3) Authors should describe how samples C1 and C2 were obtained, or at least provide a reference that describes how samples were obtained.

Ad 3) The method of preparing the samples was described in detail in [45], where similarly prepared samples were used in the chloride extraction test.

4) Describe how the compressive strength, density, and porosity tests were conducted, as well as the standard used.

Ad 4) Compressive strength, density, and porosity tests were described in the work [44] where, among others, C1 and C2 concrete were used in diffusion tests.

5) Describe the process by which the authors determined the chemical composition of the two samples. References should be provided.

Ad 5) The detailed chemical composition and basic properties of the cements are given according to the producer’s specification.

6) On page 7, should not Figure 1a be Figure 2a? This is due to the fact that Figure 1 was mentioned on page 3. It is necessary to renumber all figures starting with Figure 1 (on page 3), both in the captions and in the text logo.

Ad 6) Figure 1 has been removed, so the numbering is now correct.

7) Are the references presented by the authors at the end of the sentence "The chloride migration process was carried out similarly to the works" really correct? Please review.

Ad 7) The references have been checked and corrected.

8) All figures will need to be improved in terms of resolution, some are of poor quality.

Ad 8) All the drawings have been corrected.

9) Is there a reason why the voltage used in method 2 (30 V) differed from that used in method 1 (18 V)? Can you provide a justification for this?

Ad 9) In method 1, a voltage of 18 V was used only in the chloride ion charging stage, while in the second electrolysis stage, the same voltage as in method 2 (30 V) was used. In the second method, the electrolysis process was applied from the beginning of the test and the chloride ion source was the solution in which the samples were soaked during the test. The differences are due to the fact that both methods can be found in the literature, and the aim of the study was a preliminary analysis to select the best method for further research.

10) A dc electric field was applied to the samples in order to conduct the tests. Wouldn't it have been more beneficial to have conducted the study under an ac electric field using the technique of impedance spectroscopy? The impedance technique has been used, for example, by Bortoletto et al. [1], to study the dissolution processes and the period following the maximum conductivity value of Portland cement pastes.

[1] https://doi.org/10.1016/j.conbuildmat.2023.133566

Ad 10)

In the paper mentioned by the reviewer [1] https://doi.org/10.1016/j.conbuildmat.2023.133566, the impedance spectroscopy method was used to determine the electrical conductivity curve from which the initial and final setting times of Portland cement slurry could be determined. In this case, the spectroscopy method proved to be a sensitive technique for monitoring the hydration of Portland cement, especially early in the hydration process.

However, in my work I use the action of the electric field a little differently and I do it in two ways.

The first way is to accelerate the penetration of negatively charged chloride ions into the interior of water-saturated concrete in order to achieve a critical concentration of chloride ions that causes corrosion at the surface of the reinforcement.

The second way is to accelerate the corrosion of the reinforcement by inducing the electrolysis process of a steel rebar placed in water-saturated concrete.

Perhaps the method mentioned in the paper [1] could additionally be used to evaluate the microstructure of a concrete specimen before testing and after crack-induced destruction. This issue, however, requires analysis of the available literature and testing on separate test pieces as well as familiarization with the method itself.

11) Is there any reason why SEM analyses were not performed to evaluate the microstructure of the samples after corrosion?

Ad 11) SEM studies of the microstructure were not carried out as the research had a preliminary character. Further studies on a larger number of samples and types of concrete will include SEM studies.

12) In the text, it is unclear why after the increase in corrosion current density for M1 there is a decrease for M2 and then the current increases again for M3 and M4. I would appreciate a more detailed explanation.

Ad 12) In migration tests, when concrete is saturated with chloride ions, we can observe that an increase in corrosion current density for M1 is followed by a decrease for M2, and then the current increases again for M3 and M4. It is difficult to explain this phenomenon unequivocally, but it may be due to the fact that the corrosion products in a given measurement (measurement of M2) seal the steel-concrete contact zone, while subsequent measurements of the current are again higher due to diffusion of the corrosion products into the concrete cover. For this reason, several measurements were carried out at certain intervals, as only one measurement would be unreliable. 

13) In order to enrich the discussion, it is necessary to compare your findings with previous studies and also with those in the literature.

Ad 13) It is difficult to compare the results obtained with any previous studies, as these are preliminary studies. It is also difficult to compare the results obtained with those obtained in the literature, as they are usually carried out using different methods (different shape of samples, type of concrete).

14) Could you please explain why the accelerated process of chloride migration in concrete tests were not conducted for samples C2 in the same way as shown in Figures 5 and 6 for samples C1?

Ad 14) Two research methods were intentionally used in order to establish an appropriate research process for further research.

15) A systematic review of the manuscript is necessary in order to eliminate errors and typos.

Ad 15) A systematic review of the manuscript was carried out to eliminate errors and typos.

Reviewer 2 Report

Comments and Suggestions for Authors

Figure 1 - the text in the figure is difficult to read. In case this figure is taken directly from another source, please make sure you receive the permissions for reproducing it.

Lines 199, 208, 607, 613, 694 - please try to avoid personal statements. Use instead an impersonal mode as this is a scientific paper.

Conclusions section - it is too long; references should be removed as they represent the summary of the findings in this work (should correlations be made with scientific literature, please move them to "Discussions" section); please be more concise and state the main findings of your work (avoid general conclusions).

Section 3 - how many specimens did the authors used for each test? It should be mentioned in the manuscript

Lines 342-343 - please do not refer to figures that are not numbered in sequence. Instead, inform the reader with respect to the section/subsection where those figures are available.

Line 539 - I think the referred figure should be 8, not 9.

Comments on the Quality of English Language

Lines 58-59 - the sentence is incomplete. Please rephrase.

Line 75 - what exactly is a lollypop specimen?

Lines 221-225 - the paragraph is really difficult to read and understand. Please try to rephrase it.

Lines 226-232 - too many "however".

Line 256 - please remove "and" from "conducted and at the..."

Line 367 - please substitute "digestion" with "consumption"

Line 420 - "products quickly corrosion"?

Lines 477-479 - please rephrase the sentence. It is difficult to understand.

Author Response

Response to Reviewer 2 Comments

Comments and Suggestions for Authors

1) Figure 1 - the text in the figure is difficult to read. In case this figure is taken directly from another source, please make sure you receive the permissions for reproducing it.

Ad 1) Figure 1 was removed due to reviewer 2's recommendation to shorten the introduction. This figure is included in publication no [5].

2) Lines 199, 208, 607, 613, 694 - please try to avoid personal statements. Use instead an impersonal mode as this is a scientific paper.

Ad 2) The mistakes have been corrected.

3) Conclusions section - it is too long; references should be removed as they represent the summary of the findings in this work (should correlations be made with scientific literature, please move them to "Discussions" section); please be more concise and state the main findings of your work (avoid general conclusions).

Ad 3) The above comments have been taken into account.

4) Section 3 - how many specimens did the authors used for each test? It should be mentioned in the manuscript

Ad 4) In the first method, four samples were used. In the second method, one sample.

5) Lines 342-343 - please do not refer to figures that are not numbered in sequence. Instead, inform the reader with respect to the section/subsection where those figures are available.

Ad 5) The numbering of the drawings has been corrected.

6) Line 539 - I think the referred figure should be 8, not 9.

Ad 6) The numbering of the drawings has been corrected.

7) Comments on the Quality of English Language:

Lines 58-59 - the sentence is incomplete. Please rephrase.

Line 75 - what exactly is a lollypop specimen?

Lines 221-225 - the paragraph is really difficult to read and understand. Please try to rephrase it.

Lines 226-232 - too many "however".

Line 256 - please remove "and" from "conducted and at the..."

Line 367 - please substitute "digestion" with "consumption"

Line 420 - "products quickly corrosion"?

Lines 477-479 - please rephrase the sentence. It is difficult to understand.

 Ad 7) Errors have been corrected.

Reviewer 3 Report

Comments and Suggestions for Authors

General Comment

The submitted manuscript presents an experimental study to investigate the influence of the rebar corrosion process on the destruction of concrete cover. Two different methods were used to accelerate the corrosion: i) injection of chloride ions into the concrete using the migration method, and ii) electrolysis process. Concrete specimens made of two different type of cement were considered (CEMI and CEM III).

After an extensive literature review on the topic, where both the significance and novelty of the study are emphasized, the authors characterize the used materials, the testing samples, the testing methodology, namely the used accelerated corrosion methods, and the testing procedures. The results from the experimental campaign are presented and discussed, in light of the principal considered variables study, which are the accelerated corrosion method and the type of cement. The obtained results aim to constitute a preliminary study for a further research aimed at determining parameters enabling the further numerical modelling of corrosion process in concrete.

The topic of the manuscript is interesting and actual, since corrosion process in concrete due to the corrosion of rebars is a complex phenomenon, and more reliable numerical and analytical models are still need to predict the durability of reinforced concrete structures for the service life. For this, experimental campaigns which aim to determine parameters to calibrate such models are much needed. The presented results from this study could be useful for future researches.

I made some comments/suggestions in order to improve the manuscript. The authors should take the comments into account and revise their manuscript.

Specific Comment 1

The manuscript must be entirely revised by a professional to improve the reading, correct typos and revise technical terms.

Specific Comment 2

Title

The title should be revised, namely “the type of concrete destruction” does not sound well.

Specific Comment 3

Abstract

The abstract is too long and must be reduced (about 200 words maximum according to the mdpi rules for authors).

Specific Comment 4

Section 1 is too long and must me very reduced. In addition, “Significance” and Novelty” should be merged in a single paragraph in the end of the Introduction section.

Specific Comment 5

All symbologies must be properly defined when they first appear throughout the manuscript.

Specific Comment 6

Along the manuscript, please avoid to repeat the same ideas, in order to shorten the document.

Specific Comment 7

Section 3

In addition to the photos of the testing procedure, drawings should be added to illustrate schematically how the acceleration processes works.

Specific Comment 8

Some figures (ex., Figure 4) or tables (ex. Table A1), seems no to be cited in the text. Please revise and correct if necessary.

Specific Comment 9

Figure 5

Please increase the size of the graphs and numbers for better readability.

Specific Comment 10

Section 5

Please end this section with a paragraph summarizing the next steps/goals of this research program.

Comments on the Quality of English Language

Please see Specific Comment 1 for the authors.

Author Response

Response to Reviewer 3 Comments

1) The manuscript must be entirely revised by a professional to improve the reading, correct typos and revise technical terms.

Ad1) The manuscript has been corrected.

2) Title

The title should be revised, namely “the type of concrete destruction” does not sound well.

Ad 2) The title has been corrected.

3) Abstract

The abstract is too long and must be reduced (about 200 words maximum according to the mdpi rules for authors).

Ad 3) Abstract has been abridged.

4) Section 1 is too long and must me very reduced. In addition, “Significance” and Novelty” should be merged in a single paragraph in the end of the Introduction section.

Ad 4) Section 1 has been abbreviated.

5) All symbologies must be properly defined when they first appear throughout the manuscript.

Ad 5) All symbols have been defined.

6) Along the manuscript, please avoid to repeat the same ideas, in order to shorten the document.

Ad 6) The document has been abridged.

7) Section 3

In addition to the photos of the testing procedure, drawings should be added to illustrate schematically how the acceleration processes works.

Ad 7) Schematic drawings of the acceleration process have been added.

8) Some figures (ex., Figure 4) or tables (ex. Table A1), seems no to be cited in the text. Please revise and correct if necessary.

Ad 8) Citation of tables and figures has been revised.

9) Figure 5

Please increase the size of the graphs and numbers for better readability.

Ad 9) The size of graphs and numbers has been increased.

10) Please end this section with a paragraph summarizing the next steps/goals of this research program.

Ad 10) Paragraph summarizing the next steps has been added.

11) Comments on the Quality of English Language:

 Please see Specific Comment 1 for the authors.

Ad 11) The manuscript has been corrected.

Round 2

Reviewer 2 Report

Comments and Suggestions for Authors

The comments and suggestions brought up during the reviewing stage were addressed / considered by the authors.

The manuscript is, in my opinion, ready to be accepted for publication.

Reviewer 3 Report

Comments and Suggestions for Authors

I received the revised version of the article with revised title “The influence of corrosion processes on the degradation of concrete cover”. The authors have improved the article according to my previous comments. Hence, I consider that the article can be accepted for publication in the present form.